# Investigating workplace bullying (WPB), intention to quit and depression among nurses in the Upper West Region of Ghana

Emmanuel Dapilah[1]*, Andrews Adjei Druye[2]

1 School of Nursing, University of Rochester, Rochester, New York, United States of America, 2 Department of Adult Health, School of Nursing and Midwifery, University of Cape Coast-Ghana, Cape Coast, Ghana

* emmanuel_dapilah@urmc.rochester.edu

## Abstract

### Background

Intention to quit among nurses is increasingly recognized as a serious predictor of voluntary turnover. Voluntary turnover on the other hand is a significant factor fueling the shortage of nurses globally which could partly be blamed on negative workplace behaviors including but not limited to workplace bullying. Even though the relationship between workplace bullying and the intention to quit has been studied extensively, little is known about these concepts among nurses in Ghana.

### Aim

The purpose of this study was to establish the relationship between workplace bullying among nurses and their intention to quit the profession in the Upper West Region of Ghana. We also determined the relationship between workplace bullying and depression among nurses.

### Methods

We employed a cross-sectional design with 323 nurses recruited through a multistage sampling technique. Data were collected using a structured questionnaire with a 98.5% (N = 318) response rate.

### Results

Initial descriptive statistics indicate that 64.4% (n = 203) of the nurses had intentions of quitting the job while 52.1% (n = 164) were depressed at various degrees based on scores on the DASS-21. Further analysis shows a positive linear relationship between WPB and intentions to quit. WPB was also correlated positively with depression among the nurses. This implies that an increased incidence of bullying at work is associated with increased intention to quit and depression among the nurses.

**Data Availability Statement:** All relevant data are within the paper and its Supporting Information files.

**Funding:** The author(s) received no specific funding for this work.

**Competing interests:** The authors have declared that no competing interests exist.

## Conclusions

With over 50% of the nurses in this study intending to quit their jobs, it would be incumbent on nurse managers and other leaders at these health facilities to reconsider the work environment, policies, and leadership to prevent actual voluntary turnover. Managers must also fashion pragmatic strategies aimed at reducing stress and promoting the health and well-being of the nurses.

## Introduction

Nurses face daunting challenges at the workplace daily. One of these challenges at the health facilities is bullying among nurses. WPB has affected many nurses and continues to affect many more physically and psychologically. WPB poses a myriad of consequences to nurses, including but not limited to depression [1] and intention to quit [2]. Also, the intention to quit is a primary determinant of voluntary turnover. Voluntary turnover, on the other hand, is fueling the global shortage of nurses amid an accelerating aging population and the burden of chronic non-communicable diseases [3].

### WPB

WPB is conceptualized as repeated behaviors perpetrated by members of an organization that are considered offensive, often escalating in intensity with a perceived intent to harm [4]. While a few activities might be obvious, others will generally be covert, for example, refusing to intercede or retaining imperative information when activities are demonstrated and required for work to be done in a protected way [5]. Several studies have found that WPB is a common occurrence among nurses where it creates a harmful and fearful environment [5, 6]. It has also been established that workers who are victims of bullying have a higher incidence of ill health and related issues compared to those not bullied [6]. Numerous factors, including the stressful nature of the act, can account for this. Reports of a rise in the cases of WPB coupled with the increased relative risk of developing depression have ignited the claim that WPB remains the most devastating psychosocial workplace exposure [7]. A report from a meta-analysis indicates that exposure to negative behaviors at work predisposed employees to psychiatric and physical complaints [8]. As can be seen, the repercussions of WPB are multi-faceted and traverse physical and mental disturbances, including occupational burnout [9], depression, traumatic stress reactions, turnover intentions [10, 11], and lowered ability to deliver safe and effective patient care [12]. Furthermore, bullying at work is associated with increased absenteeism, decreased job satisfaction, an increase in medical errors, and increased job turnover [4, 13].

Also, bullying is one of the negative behaviors that has been shown to disrupt peaceful co-existence between coworkers and mar the relationships at work [14]. The interpersonal relationship between nurses and patients and the relationship required for therapeutic communication is also strained, resulting in poor health outcomes and quality of care. Also, when trust is eroded in a bullying workplace, the facility bears the brunt as employees will be reluctant to highlight or report pitfalls that occur during work.

### WPB, intention to quit, and voluntary turnover among nurses

Intention to quit and turnover are two related yet different phenomena. The intention to quit is conceptualized as an employee's voluntary willingness to leave an organization [15, 16],

while turnover, on the other hand, is mainly described as a voluntary act by nurses to leave a particular post, department, organization, or the nursing profession entirely [17]. Turnover, therefore, represents workers exiting their chosen professions for other jobs. Though these concepts differ, intention to quit is the number one predictor of voluntary turnover [18, 19]. In the view of Cho and Lewis [20], turnover intention is essential for two main reasons: it is a predictor of actual staff turnover and a signal that employees might not contribute to an organization at their full potential. The fact that an employee has decided to quit a job comes with much psychological strain, which tends to reduce his interest in the job and overall work output. Furthermore, the intention to quit is one of the most critical factors that is negatively associated with job satisfaction [21, 22], which impacts the quality of patient care. Several reasons were identified as responsible for intentions to quit including age, years of nursing experience, length of time in one's position, and career barriers [23, 24]. Also, younger age, work-related health issues, uncertainty about career prospects, and remuneration were some of the factors identified as predicting the intention to quit among nurses [25].

Staff turnover affects organizations negatively and is fueling the shortage of nurses [26, 27] and affects nurses' well-being, which in turn impacts patient care quality [27]. Voluntary turnover is of paramount importance to facilities providing healthcare globally and is considered a serious problem that impacts hospitals, wards, and individual providers [28]. The shortage brought about by turnover may result in increased rates of hospital-acquired infections, mortality, patient falls, pressure ulcers, prolonged hospital length of stay, and increased total healthcare expenditures [29].

It is worth noting that people intending to quit represent a significant precursor to them leaving the profession. Since the intention to quit is an early indicator of actual turnover [30], measures must be instituted to curb its occurrence in health facilities to reduce nurse turnover and subsequent nurse shortages. There are also suggestions that voluntary turnover increases the healthcare expenditure of most facilities [29], placing a substantial financial burden on the affected institutions, negatively influencing staff incentives, and compromising the quality of care amid limited resources. Several researchers have studied the association between WPB and the intention to quit and have found a positive linear association between these variables [31–35].

## WPB, depression, intention to quit, and selected demographic factors

WPB, intention to quit, and depression can be influenced by several factors, including age, sex, professional rank, and category within the nursing profession. However, the available literature regarding the characteristics of the risk groups that are likely to be victims of WPB is inconsistent and inconclusive [36]. People can be discriminated against in the workplace due to their age, and this may lead to negative attitudes and practices carried out in the workplace [37]. Regarding the association between age and WPB, the results were mixed. Several studies have found no significant differences in the prevalence of WPB among the various age groups [36, 38, 39], while Feijó and colleagues specifically concluded that workers younger than 44 years old were more likely to be bullied [40]. Additionally, age and years of experience have been identified as responsible for the intention to quit in some studies [23, 41]. Both men and women may experience bullying at work differently. On gender differences in the prevalence of depression, the literature shows a significant variation between males and females. In a large multinational study to examine depression among men and women aged 18–75 in 23 European countries, researchers found that women reported higher levels of depression than men in all countries [42]. Research has also shown that younger males demonstrate a much higher intent to leave the nursing profession than their colleagues [43].

The relationship between WPB and the ranks of nurses is a critical subject of focus since nurses within different positions in the profession experience bullying differently. It is suggested that individuals in positions of power use their authority to intimidate or undermine their subordinates [44], fueling concerns that nurses within the lower ranks may report more bullying episodes, depression, and higher intentions to quit the profession. For example, Al-Ghabeesh and Qattom [38] found that nurses with less experience in the emergency department were exposed to more bullying compared to other nurses. Therefore, we believe junior nurses may be bullied more, be more depressed, and show higher intentions to quit than nurses in the higher ranks. In Ghana, Enrolled Nurses (EN) and Registered General Nurses (RGN) differ in their educational backgrounds. ENs typically undergo a 2-year certificate program with an emphasis on practical skills heralded by a focus on hands-on training in clinical settings. Conversely, the RGNs pursue an advanced program either at the diploma level or degree level with a more comprehensive curriculum. Subsequently, the RGNs are placed in a higher position in the workplace with more responsibilities and oversight than the ENs. Hence, the ENs are in a disadvantaged position and might be subjected to higher levels of bullying, depression, and the intention to quit the profession.

## Relationship between WPB and depression among nurses

Depression is a mental state characterized by feelings of sadness, loss of interest or pleasure, and feelings of worthlessness or hopelessness [45]. Over the years, depression and anxiety disorders have been identified as common mental ailments that pose dangers to individuals and society [46]. In a systematic review and meta-analysis, Xie, Qin [47] found that about 44% of Chinese nurses had depressive symptoms. In a survey conducted by Hansson, Chotai [48], 33% of the patients with mood disorders attributed their mental problems to circumstances at the workplace, placing issues in the work environment as the number one cause of depression among employees. In a recent survey, Fond, Fernandes [49] found about 30% of health workers had met the criteria for clinical depression, with WPB identified as one of the significant risk factors for depression.

The fact that work is integral to employees' psychological health is not far-fetched, as they spend many hours daily at work. Work or employment is a two-edged sword. While it offers remuneration, meaning, and avenues for social interactions, it can also be a source of great stress [50]. Apart from work posing as a source of stress to the worker, other work-related factors and harmful workplace social acts can poison the work environment, thereby worsening the situation. Among the many work-related factors implicated in workers' mental health problems is the WPB phenomenon. On an individual level, WPB affects all aspects of a worker's being and might result in health-related issues such as severe headaches, depression, and anorexia [51]. In a study investigating WPB and depression among physicians, it was established that instances of bullying at work correlated positively with the risk of developing depression [52]. Furthermore, several other studies have found a positive linear relationship between WPB and depression among nurses [49, 53, 54].

According to the literature, WPB could lead to intention to quit and depression among nurses. The intention to quit and depression among nurses compromise their health and the quality of work output and patient care. However, there is scanty research on these concepts in most developing countries, including Ghana. This study aimed to determine the prevalence of WPB and depression and establish the relationship between WPB, depression, and the intention to quit. It sought to answer the following specific research questions:

1. What is the prevalence of depression and intention to quit among nurses working in the Upper West Region of Ghana?

2. What is the relationship between WPB, depression, and the intention to quit among the nurses?

It is essential to study nurses' intentions to quit so that retention strategies can be drafted and implemented to reduce and/or prevent voluntary turnover.

## Methods

### Study design

We conducted this study using a descriptive cross-sectional design and a non-experimental quantitative approach. This design was chosen because data was collected at one point [55] primarily to determine prevalence and establish associations between the study variables [56]. This study was conducted among 323 RGN and EN working in five public hospitals in the Upper West Region (UWR) of Ghana. The selection of study respondents was done through a stratified sampling technique, a probability method. Our choice of this technique was informed by the fact that it would have been challenging to track nurses across the region individually. The multistage sampling was conducted through the following steps. Firstly, we calculated the sample size using Yamane [57] sample size calculation formula. Secondly, we selected the Districts from the Region to be included. Eight (8) districts were included in the study because three did not have a hospital at the time of the study. Thirdly, hospitals in these eight (8) districts were used as study sites. Finally, using proportions, a simple random sampling technique was used to select nurses from each of the eight district hospitals. The nurses were divided into two groups based on their professional background: EN and RGN. The final sample consisted of 46.7% (n = 151) RGN (84 males and 67 females) and 53.3% (n = 172) EN (83 males and 89 females), for a total of 323 nurses (167 males and 156 females).

### Data collection instruments

We collected the data using a structured self-administered questionnaire. A questionnaire was used to collect the data because, according to Leavy [55], questionnaires are less exorbitant and require less time and energy to administer, ensure anonymity or, to some extent, perceived privacy, and minimize response bias. This questionnaire has four sections: demographic variables, intention to quit, WPB, and depression among nurses.

**Demographic variables.** The demographic variables of the respondents collected in this study include gender, age, professional background, and rank/position in the profession. These are seen as covariates in the relationship between WPB, depression, and the intention to quit, so they were controlled during the multivariate analysis.

**Prevalence of WPB.** We adopted the Negative Acts Questionnaire-Revised (NAQ-R) developed by Einarsen, Hoel [58] to measure bullying. Each of the 22 items on the instrument was rated on a 5-point Likert scale (1 = never, 2 = now and then, 3 = monthly, 4 = weekly, 5 = daily). Total scores on bullying ranged from 22 to 110 points. Concerning current developments on the use of cutoff criteria in determining targets of WPB; respondents with a cutoff score less than 33 were classified as not bullied. Respondents with scores between 33 and 44 were considered occasionally bullied or targets of WPB. From a score of 45 upwards, respondents were considered victims of WPB [59]. The 22 items on the instrument have categorized bullying into three types: **person-related bullying, work-related bullying, and physical intimidation.** A 23rd question, which is a single item that measures self-reported bullying at work in the past 6 months, is preceded by a generally accepted definition of bullying with several questions about the bullying experience, such as frequency, duration, and the principal bullies [58].

The scale's reliability, as tested by Cronbach's alpha in this study, was 0.929, indicating excellent internal consistency (S1 File).

**Level of depression among nurses.** The level of depression was measured with Depression Anxiety Stress Scale version 21 (DASS-21), which is a short form of DASS-42 developed by Lovibond [60] to assess the core symptoms of depression, anxiety, and stress. The DASS-21 is made up of 21 questions which are divided into three categories (Depression, Anxiety, and Stress) of seven items, each measured over the past week, and scores range from 0, "Did not apply to me at all," to 3, "Applied to me very much, or most of the time." The Depression sub-scale looks at hopelessness, low self-esteem, and low positive affect. The Anxiety sub-scale measures autonomic arousal, physiological hyperarousal, and the subjective feeling of fear. The Stress sub-scale items consider tension, agitation, and negative affect. For each subscale, the scores for the identified items are summed. Because DASS-21 is a short version of the original 42-item scale, the final score of each subscale (Depression, Anxiety, and Stress) was multiplied by two (x2) and then compared with the DASS Severity Ratings (see Table 1).

Cronbach's alpha for the DASS-21 and depression subscales for this study were 0.85 and 0.74, respectively.

**Intention to quit.** Finally, intention to leave (quit) was measured with a single-item, five-point scale constructed by Einarsen, Hoel [58]. The respondents were required to answer the question: *"Have you considered quitting your present job over the last six months?"* based on responses that ranged from 0 *(Never)* to 4 *(Very often)*.

## Ethical considerations

Ethical clearance was obtained from the University's Institutional Review Board, and permission was sought from the management of the various hospitals from which the participants were recruited. Furthermore, all the participants provided written informed consent by endorsing the consent form attached to the questionnaire before answering the questions. The research assistants witnessed the signing of the consent forms. Ethical considerations stipulated by the Helsinki Declarations were also strictly adhered to.

## Data collection procedures

We desired to maximize the response rate amid time and budgetary constraints. Therefore, considering many factors, the in-person delivery method was adopted for this study. Leavy [55] posits that in-person surveys generally occur in group settings and have the highest response rate. Two assistants were trained to help with the in-person distribution and subsequent collection of the questionnaires, which spanned from 25th February 2020 to 17th April 2020. A total of 323 questionnaires were distributed across the eight study sites. However, we retrieved 318 questionnaires representing a response rate of 98.5%. Also, three (3)

**Table 1. DASS severity ratings.**

| Severity | Depression | Anxiety | Stress |
|---|---|---|---|
| Normal | 0–9 | 0–7 | 0–14 |
| Mild | 10–13 | 8–9 | 15–18 |
| Moderate | 14–20 | 10–14 | 19–25 |
| Severe | 21–27 | 15–19 | 26–33 |
| Extremely severe | 28+ | 20+ | 34+ |

Source: Lovibond (1995).

questionnaires from the 318 with incomplete entries representing 0.9% were rejected. Hence, the final analysis was conducted with data from 315 questionnaires.

## Data analysis

Data analysis procedures allow us to determine the answers to the research questions or hypotheses [55]. The data analysis was performed in SPSS version 29, and the alpha level was set at 0.05. Initially, frequencies were run for the data, and inspection was done to identify missing values and errors. Some ordinal variables were transformed into dichotomous variables to allow for better result presentation, discussion, and communication of findings. For example, the variable 'age' collected as a continuous variable was transformed into a categorical variable based on the following groups: 20–29, 30–39, and 40+ years. Current professional ranks were grouped into two: junior nurses and senior nurses. All items in the questionnaire measuring WPB were transformed into a single variable to facilitate data analysis. As such, the 22 items in the NAQ-R were coded and transformed into one main variable, which measured exposure to bullying. The first part of the NAQ-R has 22 items, and each item was measured on a 5-point Likert scale (1-never, 2-now and then, 3-monthly, 4-weekly, and 5-daily). The total score on the NAQ-R was, therefore, 110 (5*22 = 110). The total score obtained by each respondent was further categorized into three groups according to cutoff points by Notelaers and Einarsen (2013) as follows: less than 33, not bullied; 33–44, occasionally bullied or targets of bullying; and 45+, victims of bullying.

Similarly, seven (7) items on the DASS-21 measured depression. Each item was measured on a 4-point Likert scale from 0 to 3. So, the total score for depression as measured by the DASS-21 was 21 (3*7 = 21). However, since the DASS-21 is an abridged version of the original DASS-42, whatever score was obtained from DASS-21 was multiplied by 2 to obtain the actual depression score. Based on the actual scores, the level of depression among the respondents was categorized according to the DASS severity rating by Lovibond [60] as follows: 0–9 = normal, 10–13 = mild depression, 14–20 = moderate depression, 21–27 = severe depression, and 28+ = extremely severe depression.

# Results

## Demographic variables of respondents

The initial analysis shows that 52.1% (n = 164) were males, while 47.9% (n = 151) were females. We observed that about 47.9% of the nurses were between the ages of 20 to 29 years. This was followed by about 39.4% for those between 30 and 39 years old. The least were those aged 40 years and above, representing only 12.7%. Regarding the professional background of the nurses, it was observed that 53% were EN, while the remaining proportion (47%) were RGN (see Table 2 for details).

## Intention of nurses to quit the profession

Regarding nurses' intention to quit, 64.4% (n = 203) have indicated that they have considered leaving the nursing profession within the past six months, while 35.6% (n = 112) said they have not contemplated leaving the profession (see Table 3).

## Prevalence of depression among nurses

Regarding depression, 52.1% (n = 164) of the nurses over the study period have experienced signs and symptoms of depression at various levels based on scores on the DASS-21 (see Table 4).

**Table 2. Demographic characteristics of respondents.**

| Demographic variables | Frequencies | Percentages (%) |
|---|---|---|
| *Gender* | | |
| Male | 164 | 52.1 |
| Female | 151 | 47.9 |
| **Total** | **315** | **100.0** |
| *Age (years)* | | |
| 20–29 | 151 | 47.9 |
| 30–39 | 124 | 39.4 |
| 40+ | 40 | 12.7 |
| **Total** | **315** | **100.0** |
| *Professional background* | | |
| Registered General Nurses | 148 | 47.0 |
| Enrolled Nurses | 167 | 53.0 |
| **Total** | **315** | **100.0** |
| *Current professional rank* | | |
| Junior Nurses | 71 | 22.5 |
| Senior Nurses | 244 | 77.5 |
| **Total** | **315** | **100.0** |

Source: Field Survey, (2020)

**Table 3. Intention of nurses to quit the profession.**

| | | Frequency | Percent |
|---|---|---|---|
| Valid | Never | 112 | 35.6 |
| | Rarely | 63 | 20.0 |
| | Sometimes | 92 | 29.2 |
| | Quite often | 32 | 10.2 |
| | Very often | 16 | 5.1 |
| | **Total** | **315** | **100.0** |

Source: Field Survey (2020).

## Relationship between WPB and the intention to quit

The relationship between WPB and the intention of a nurse to quit his/her job was also assessed, and the results are displayed in Table 5.

**Table 4. Prevalence of depression among nurses based on DASS-21.**

| | | Frequency | Percent |
|---|---|---|---|
| Valid | Normal | 151 | 47.9 |
| | Mild depression | 84 | 26.7 |
| | Moderate depression | 59 | 18.7 |
| | Severe depression | 16 | 5.1 |
| | Extremely severe depression | 5 | 1.6 |
| | Total | 315 | 100.0 |

Source: Field Survey (2020).

**Table 5. Correlations of WPB and intentions to quit.**

| | | Intentions of Respondents to Quit Current Job |
|---|---|---|
| Total perceived bullying | Pearson Correlation | 0.487** |
| | Sig. (2-tailed) | 0.000 |
| | **N** | **315** |

Source: Field Survey (2020) **Significant at p<0.001

The relationship between perceived bullying at work (as measured by the NAQ-R) and nurses' intention to quit the profession (measured by a single item on a 5-point Likert scale) was investigated using the Pearson product-moment correlation coefficient. Initial investigations were carried out to rule out violations of the assumptions of normality, linearity, and homoscedasticity. The results show a strong, positive correlation between the two variables (r = 0.487; N = 315, $p<0.001$), with high perceived WPB associated with increasing nurses' intentions to quit their jobs. This means that as the perceived level of WPB increases, the level of the intention of a nurse to quit his/her job increases and vice versa.

## Relationship between intention to quit and depression among nurses

We assessed the relationship between the intention of a nurse to quit his/her job and depression using Pearson correlation (see Table 6 for details).

The results indicate a strong, positive correlation between the two variables (r = 0.363; N = 315, $p<0.001$), with high levels of depression associated with increasing levels of the intentions of nurses to quit their jobs. This means that as the perceived level of depression increases, the level of the intention of a nurse to quit his/her job increases and vice versa.

## Relationship between WPB and depression among nurses

We also determined the relationship between WPB and depression among the respondents. Again, the relationship between perceived bullying at work (as measured by the NAQ-R) and depression among nurses (as measured by the DASS-21) was investigated using the Pearson product-moment correlation coefficient. Initial investigations were carried out to rule out violations of the assumptions of normality, linearity, and homoscedasticity, and the results are presented in Table 7.

From the analysis, there was a strong, positive correlation between the two variables (r = 0.559; N = 315; $p<0.001$), with high levels of perceived WPB being associated with increasing levels of depression among nurses. Thus, the higher the perceived level of WPB, the higher the level of depression among the nurses and vice versa.

To find out if WPB and depression were significant independent predictors of nurses' intention to quit their jobs, we conducted multiple linear regression analyses. All the assumptions of the general linear models were not violated. A unit increase in WPB leads to a 0.036

**Table 6. Relationship between intention to quit and depression among nurses.**

| | | Actual depression score |
|---|---|---|
| Intention to Quit | Pearson Correlation | 0.363** |
| | Sig. (2-tailed) | 0.000 |
| | **N** | **315** |

Source: Field Survey (2020) **Significant at p<0.001

**Table 7. Relationship between WPB and depression.**

|  |  | Actual depression score |
|---|---|---|
| Total perceived bullying | Pearson Correlation | 0.559** |
|  | Sig. (2-tailed) | 0.000 |
|  | **N** | **315** |

Source: Field Survey (2020) **Significant at p<0.001

unit increase in the intention to quit while controlling for depression, age, professional background, gender, and rank of the nurse (B = 0.036, 95% CI 0.025; 0.046). We can be 95% confident that the true population parameter lies between 0.025 and 0.046. Also, a unit increase in depression leads to a 0.026 unit increase in the intention to quit while controlling for WPB, age, professional background, gender, and rank of the nurse (B = 0.026, 95% CI 0.005; 0.048). Hence, we can be 95% confident that the true population parameter lies between 0.005 and 0.048. The adjusted $R^2$ for the regression model is 0.244. This indicates that 24.4% of the variance in the nurses' intention to quit is accounted for by WPB and depression. (see Table 8 for details).

## Discussion

More than 60% of the nurses in this study have stated that they have contemplated leaving the profession within the past six months. This figure is enormous because most of the facilities in Ghana have a shortfall in the number of nurses. It was necessary to ascertain whether WPB was related to the nurses' intentions to quit their jobs. The Pearson correlation coefficient indicated that WPB was positively related to nurses' intention to quit the profession. This means increased WPB prevalence could increase nurses' intention to leave their profession and vice versa. Several studies have also established a positive association between WPB and nurses' intentions to leave the profession [4, 31, 32, 35, 61].

Intention to quit could result in actual staff turnover. Staff turnover leads to the loss of qualified nurses, and according to Bae, Cho [62] leads to compromised care outcomes and high operational expenses [29]. Nurses who are victims of WPB have increased chances of falling sick and experiencing other health-related issues than workers who are not bullied [6]. Because WPB is prevalent among nurses, it stands to reason that bullying will pose health and safety risks to them, which may ultimately trigger their intentions to leave the profession for other

**Table 8. Summary of multiple linear regression.**

|  |  |  |  |  |  | 95% CI for B | |
|---|---|---|---|---|---|---|---|
|  | **B** | **SE** | **Beta** | **t** | **P** | **Lower bound** | **Upper bound** |
| WPB | 0.036 | 0.005 | 0.400 | 6.719 | <0.001 | 0.025 | 0.046 |
| Depression | 0.026 | 0.011 | 0.142 | 2.374 | 0.018 | 0.005 | 0.048 |
| 20–29 years (REF) |  |  |  |  |  |  |  |
| 30–39 years | 0.143 | 0.132 | 0.059 | 1.088 | 0.277 | -0.116 | 0.402 |
| 40+ years | 0.316 | 0.203 | 0.088 | 1.559 | 0.120 | -0.083 | 0.715 |
| Gender | -0.065 | 0.118 | -0.027 | -0.553 | 0.581 | -0.297 | 0.167 |
| Professional Nursing Background | 0.004 | 0.144 | 0.002 | 0.030 | 0.976 | -0.278 | 0.287 |
| Professional rank/position | 0.099 | 0.185 | 0.035 | 0.537 | 0.592 | -0.265 | 0.464 |

a. Dependent Variable: Intentions of Respondents to Quit Current Job

jobs that are less stressful and risky. This might help explain the positive association between WPB and the nurses' intentions to quit the profession.

In this study, over 50% of the nurses have experienced various degrees of depression according to measures on the DASS-21. However, it is important to note that DASS-21 is a quantitative measure of distress (depression, anxiety, and stress) and therefore not a definite measure of clinical diagnosis [60]. What this means is that the assessment scores obtained from the DASS-21 can be used to identify individuals experiencing considerable symptoms who might be at high risk of developing clinical depression. There is, therefore, a need to refer such people for further psychological evaluation.

Further investigations were conducted on the relationship between WPB and depression among nurses who took part in the study. It was determined from the Pearson correlation coefficient that there was a strong positive linear relationship between WPB and depression among the nurses. This is an indication that any increase in WPB prevalence subsequently would be associated with an increase in the nurses' depression states and vice versa. This finding is consistent with earlier studies [2, 49, 53, 54], which found a positive linear relationship between WPB and depression.

Work offers livelihood for employees and, at the same time, can be seen as a source of great stress. According to previous studies, WPB has been associated with decreased mental health [54, 63]. As such, anything that negatively affects work or the working environment will inadvertently increase employees' stress levels and result in compromised well-being. This might account for the positive linear relationship between WPB and depression. However, the findings in this study contradict those of Doe [64], who found no significant association between WPB and depression among a sample of university workers in Ghana. The reasons for the difference in findings are not readily known. Hence, further research is required to help elucidate these findings.

The multiple linear regression results have indicated that WPB (B = 0.036, 95% CI = 0.025; 0.046) and depression (B = 0.026, 95% CI = 0.005; 0.048) are independent predictors of nurses' intention to quit their jobs even after adjusting for the nurses' age, gender, professional background, and position. The whole regression model accounted for 24.4% of the variance in the intention to quit. This means that although WPB and depression are significant factors in the decision of nurses to quit their profession, there are other reasons not captured by this study. We can speculate that low salaries and remunerations, low prestige, discrimination, and negative public perceptions about the nurse could be accountable for nurses intending to quit their jobs in Ghana. These are similar to the reasons identified by Enea, Maniscalco [25], who found work-related health issues, uncertainty about career prospects, and remuneration as precursors to the decision to quit. The intention of workers to quit their professions has been found to predict actual staff turnover strongly, and this indicator has been used widely in nurse turnover studies [18, 19]. However, with the high unemployment levels among newly qualified nurses and the unavailability of alternative job opportunities for nurses in Ghana, we are uncertain whether the intention to quit would result in actual nurse turnover. Therefore, longitudinal studies need to be conducted to determine if the intention to quit among nurses will lead to staff turnover.

## Conclusions

WPB, intention to quit, and depression are prevalent among nurses in this study area. There is evidence that a substantial number of nurses in this study are considering quitting their current job as nurses, and most of them are depressed. The intention to quit among the nurses was found to be predicted by bullying at the facilities and depression among the nurses. Once

it is established that WPB and depression are independent predictors of the nurses' intention to quit, measures must be instituted to curb the occurrence of bullying at work and depression among the nurses. When there is a reduction in the incidence of bullying at work, the chances that workers will desire to quit their jobs or become depressed might be minimized. This will create a cordial and productive work environment, ultimately leading to quality care outcomes and a healthy hospital workforce.

## Recommendation for further research

Considering the number of nurses in this study who have the intention to quit their jobs amidst the high unemployment among newly qualified nurses and the lack of alternate job opportunities for nurses in Ghana, we recommend that researchers carry out longitudinal studies to establish whether the intention to quit one's job will result in actual staff turnover. There is also a need for studies that will comprehensively study the factors responsible for nurses intending to leave the profession since the current study accounted for only 24% of the variance in the intention to quit.

## Supporting information

**S1 File. Sample questionnaire.**
(DOCX)

**S2 File. Dataset.**
(SAV)

## Acknowledgments

The corresponding author received financial support from the Samuel and Emelia Brew-Butler-SGS/GRASAG UCC Research Fund 2020 while preparing the thesis, which was submitted to the University of Cape Coast in partial fulfillment of the requirements for the award of a Master of Nursing (MN).

## Author Contributions

**Conceptualization:** Emmanuel Dapilah.

**Formal analysis:** Emmanuel Dapilah.

**Investigation:** Emmanuel Dapilah.

**Methodology:** Emmanuel Dapilah.

**Supervision:** Andrews Adjei Druye.

**Validation:** Andrews Adjei Druye.

**Writing – original draft:** Emmanuel Dapilah.

**Writing – review & editing:** Emmanuel Dapilah, Andrews Adjei Druye.

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
