## [Decision Letter · Decision Letter 0]

28 Aug 2024

PONE-D-24-20237Investigating Workplace Bullying, Intention to Quit and Depression Among Nurses in the Upper West Region of Ghana.PLOS ONE

Dear Dr. Dapilah,

Thank you for submitting your manuscript to PLOS ONE. After careful consideration, we feel that it has merit but does not fully meet PLOS ONE’s publication criteria as it currently stands. Therefore, we invite you to submit a revised version of the manuscript that addresses the points raised during the review process.

We look forward to receiving your revised manuscript.

Kind regards,

Francesco Marcatto, Ph.D.

Academic Editor

PLOS ONE

Journal Requirements:

2. Please amend your authorship list in your manuscript file to include author Emmanuel Dapilah and Andrews Adjei Druye.

3. Please amend your manuscript to include your abstract after the title page.

Reviewers' comments:

Reviewer's Responses to Questions

**Comments to the Author**

1. Is the manuscript technically sound, and do the data support the conclusions?

Reviewer #1: Yes

Reviewer #2: Partly

2. Has the statistical analysis been performed appropriately and rigorously? 

Reviewer #1: Yes

Reviewer #2: Yes

3. Have the authors made all data underlying the findings in their manuscript fully available?

Reviewer #1: Yes

Reviewer #2: Yes

4. Is the manuscript presented in an intelligible fashion and written in standard English?

Reviewer #1: Yes

Reviewer #2: Yes

5. Review Comments to the Author

Reviewer #1: Dear authors,

Thank you for providing me with the opportunity to review this manuscript. It is well-written. However, there are some minor changes that I recommend making:

1. In the abstract, you mentioned using a multistage sampling technique, but in the methods section, you stated that a cluster sampling technique was used. Please ensure consistency in your approach.

2. In the data analysis section, you mentioned that the analysis was conducted to address research questions and hypotheses. However, there were no research questions or hypotheses presented in the manuscript.

3. Many studies cited in the literature review are outdated. Please provide justification for using older studies or update the literature review with more recent studies.

Thank you for your attention to these suggestions.

Reviewer #2: I have read with interest the manuscript titled ‘Investigating Workplace Bullying, Intention to Quit and Depression Among Nurses in the Upper West Region of Ghana’, which presents a correlational study testing the relationship between workplace bullying and turnover intention and depression among nurses in Ghana.

I appreciated the rationales behind the research, indeed, replicating previous finding and extending them to diverse social and cultural contexts is important; however, I have some concerns about the current form of the manuscript, which I will outline below.

My concerns are mainly focused on the rationales to include/exclude factors in the major statistical analyses and their theoretical importance. Indeed, I generally found that in their design and data analyses authors took into consideration some factors (e.g., registered general nurses vs. enrolled nurses, gender, age groups, rank) without these factors being necessarily introduced in the theoretical introduction. It is unclear what the literature says about the importance of these factors in the relationship between bullying and intentions to quit work and or depression among nurses.

So, I found an inconsistency between some analyses (e.g., multiple linear regression) and theoretical background.

I suggest that the authors explain in the theoretical introduction why these factors are important and explain why they were taken into consideration based on the literature of reference.

As an example, as a reader I was (and still I am) unaware about the difference between registered general nurses and enrolled nurses. I think it is necessary to explain what the difference is and why this factor was taken into consideration. Is it important for the relationship between workplace bullying and the outcome variables? Or do the authors considered it in an exploratory fashion. The same rationales apply to age group, gender and rank position. Moreover, why the authors decided to collapse those specific age groups? What was the rationale? Why not to include age as a continuous variable if age is considered an important factor?

Another point, on page 6 of the manuscript, the authors indicate that the questionnaire used can be found in Appendix A. However, downloading the supplementary materials, one does not find the questionnaire with the measures used, but other materials (e.g., Ethical clearance and Consent Form). While I found that, in the section describing the measures, there were enough information about how depression and intention to quit work were present, I suggest the authors to provide more information on the measure of workplace bullying. For example, does the NAQ-R tap into a single dimension or into several ones? If it has subscales, which dimensions they represents and why they were collapse into a single variable as specified at page 9 of the current manuscript. Also, I found it would be interesting for the reader to have examples of items if the questionnaire is not available as an appendix.

6. PLOS authors have the option to publish the peer review history of their article (what does this mean?). If published, this will include your full peer review and any attached files.

Reviewer #1: No

Reviewer #2: No

---

## [Author Response · Author response to Decision Letter 0]

16 Sep 2024

Response to reviewers’ comments

Reviewer #1: Dear authors,

Thank you for providing me with the opportunity to review this manuscript. It is well-written. However, there are some minor changes that I recommend making:

1. In the abstract, you mentioned using a multistage sampling technique, but in the methods section, you stated that a cluster sampling technique was used. Please ensure consistency in your approach.

Response: Thank you for this observation. We have edited this portion to reflect the exact sampling technique i.e. stratified sampling and ensured its consistent usage across the manuscript.

2. In the data analysis section, you mentioned that the analysis was conducted to address research questions and hypotheses. However, there were no research questions or hypotheses presented in the manuscript.

Response: we meant the analysis was done to address the purpose of the study. We have revised the manuscript to correct this. Thank you.

3. Many studies cited in the literature review are outdated. Please provide justification for using older studies or update the literature review with more recent studies.

Response: we have replaced some of the old references where possible with current ones. 

Thank you for your attention to these suggestions.

Reviewer #2: I have read with interest the manuscript titled ‘Investigating Workplace Bullying, Intention to Quit and Depression Among Nurses in the Upper West Region of Ghana’, which presents a correlational study testing the relationship between workplace bullying and turnover intention and depression among nurses in Ghana.

I appreciated the rationales behind the research, indeed, replicating previous finding and extending them to diverse social and cultural contexts is important; however, I have some concerns about the current form of the manuscript, which I will outline below.

My concerns are mainly focused on the rationales to include/exclude factors in the major statistical analyses and their theoretical importance. Indeed, I generally found that in their design and data analyses authors took into consideration some factors (e.g., registered general nurses vs. enrolled nurses, gender, age groups, rank) without these factors being necessarily introduced in the theoretical introduction. It is unclear what the literature says about the importance of these factors in the relationship between bullying and intentions to quit work and or depression among nurses.

So, I found an inconsistency between some analyses (e.g., multiple linear regression) and theoretical background.

I suggest that the authors explain in the theoretical introduction why these factors are important and explain why they were taken into consideration based on the literature of reference.

As an example, as a reader I was (and still I am) unaware about the difference between registered general nurses and enrolled nurses. I think it is necessary to explain what the difference is and why this factor was taken into consideration. Is it important for the relationship between workplace bullying and the outcome variables? Or do the authors considered it in an exploratory fashion. The same rationales apply to age group, gender and rank position. Moreover, why the authors decided to collapse those specific age groups? What was the rationale? Why not to include age as a continuous variable if age is considered an important factor?

Response: we appreciate your concerns. We have included some literature regarding the demographic factors. Though the demographic variables may play significant role in the relationship between WPB, depression, and intention to quit, they are not the primary factors in this study and only served as covariates. Also, the variable age was categorized into groups to ascertain if certain age groups will be more likely to be affected than others. For example, the literature shows that younger nurses and junior nurses have a higher propensity to leave the profession and to be bullied.

Another point, on page 6 of the manuscript, the authors indicate that the questionnaire used can be found in Appendix A. However, downloading the supplementary materials, one does not find the questionnaire with the measures used, but other materials (e.g., Ethical clearance and Consent Form). While I found that, in the section describing the measures, there were enough information about how depression and intention to quit work were present, I suggest the authors to provide more information on the measure of workplace bullying. For example, does the NAQ-R tap into a single dimension or into several ones? If it has subscales, which dimensions they represents and why they were collapse into a single variable as specified at page 9 of the current manuscript. Also, I found it would be interesting for the reader to have examples of items if the questionnaire is not available as an appendix.

Response: we have realized we did not attach the questionnaire as an appendix as indicated in the manuscript. Hence, we have attached the questionnaire in this revision as S1 Appendix A. Also, we have included adequate information on the measure of workplace bullying as suggested. Thank you for your input.

---

## [Editor Report · Decision Letter 1]

24 Sep 2024

PONE-D-24-20237R1Investigating workplace bullying, intention to quit and depression among nurses in the Upper West Region of Ghana.PLOS ONE

Dear Dr. Dapilah,

Thank you for submitting your manuscript to PLOS ONE. After careful consideration, we feel that it has merit but does not fully meet PLOS ONE’s publication criteria as it currently stands. Therefore, we invite you to submit a revised version of the manuscript that addresses the points raised during the review process.

Dear Dr. Dapilah,

Thank you for submitting the revised version of your manuscript. While the cover letter indicates that the paper has been updated according to the reviewers' comments, I was unable to find any changes addressing the feedback from Reviewer 1. Specifically, this includes:

The description of the sampling technique, both in the abstract and the main text;The sentence about the research questions and hypotheses;The use of outdated literature.

I kindly ask you to submit an updated version of your manuscript that incorporates these changes. Additionally, the acronym **WPB** is used in the text but is never explained. Please address this as well.

We look forward to receiving your revised manuscript.

Kind regards,

Francesco Marcatto, Ph.D.

Academic Editor

PLOS ONE
---

## [Author Response · Author response to Decision Letter 1]

8 Oct 2024

Response to reviewers’ comments

Reviewer #1: Dear authors,

Thank you for providing me with the opportunity to review this manuscript. It is well-written. However, there are some minor changes that I recommend making:

1. In the abstract, you mentioned using a multistage sampling technique, but in the methods section, you stated that a cluster sampling technique was used. Please ensure consistency in your approach.

Response: Thank you for this observation. We have edited this portion to reflect the exact sampling technique and stratified sampling and ensured its consistent usage across the manuscript. The sampling technique is mentioned in the abstract and described in the main text. A supplementary table of the nurses recruited from each of the eight hospitals has been added. 

2. In the data analysis section, you mentioned that the analysis was conducted to address research questions and hypotheses. However, there were no research questions or hypotheses presented in the manuscript.

Response: Thank you for this comment. We meant the analysis was done to address the study's purpose. We have revised the manuscript to correct this. We have stated the study's purpose and added the specific questions that it sought to answer. 

3. Many studies cited in the literature review are outdated. Please provide justification for using older studies or update the literature review with more recent studies.

Response: We appreciate your concerns about using outdated literature and have replaced several. Most of the citations are now within the last 10 years. However, few foundational studies have been cited since we needed to cite the original authors. 

Thank you for your attention to these suggestions.

Reviewer #2: I have read with interest the manuscript titled ‘Investigating Workplace Bullying, Intention to Quit and Depression Among Nurses in the Upper West Region of Ghana’, which presents a correlational study testing the relationship between workplace bullying and turnover intention and depression among nurses in Ghana.

I appreciated the rationales behind the research, indeed, replicating previous finding and extending them to diverse social and cultural contexts is important; however, I have some concerns about the current form of the manuscript, which I will outline below.

My concerns are mainly focused on the rationales to include/exclude factors in the major statistical analyses and their theoretical importance. Indeed, I generally found that in their design and data analyses authors took into consideration some factors (e.g., registered general nurses vs. enrolled nurses, gender, age groups, rank) without these factors being necessarily introduced in the theoretical introduction. It is unclear what the literature says about the importance of these factors in the relationship between bullying and intentions to quit work and or depression among nurses.

So, I found an inconsistency between some analyses (e.g., multiple linear regression) and theoretical background.

I suggest that the authors explain in the theoretical introduction why these factors are important and explain why they were taken into consideration based on the literature of reference.

As an example, as a reader I was (and still I am) unaware about the difference between registered general nurses and enrolled nurses. I think it is necessary to explain what the difference is and why this factor was taken into consideration. Is it important for the relationship between workplace bullying and the outcome variables? Or do the authors considered it in an exploratory fashion. The same rationales apply to age group, gender and rank position. Moreover, why the authors decided to collapse those specific age groups? What was the rationale? Why not to include age as a continuous variable if age is considered an important factor?

Response: we appreciate your concerns. We have included some literature regarding the demographic factors. Though the demographic variables may play a significant role in the relationship between WPB, depression, and intention to quit, they are not the primary factors in this study and only served as covariates. Also, the variable age was categorized into groups to ascertain if certain age groups would be more likely to be affected than others. For example, the literature shows that younger and junior nurses have a higher propensity to leave the profession and be bullied.

Another point, on page 6 of the manuscript, the authors indicate that the questionnaire used can be found in Appendix A. However, downloading the supplementary materials, one does not find the questionnaire with the measures used, but other materials (e.g., Ethical clearance and Consent Form). While I found that, in the section describing the measures, there were enough information about how depression and intention to quit work were present, I suggest the authors to provide more information on the measure of workplace bullying. For example, does the NAQ-R tap into a single dimension or into several ones? If it has subscales, which dimensions they represents and why they were collapse into a single variable as specified at page 9 of the current manuscript. Also, I found it would be interesting for the reader to have examples of items if the questionnaire is not available as an appendix.

Response: We realized we did not attach the questionnaire as an appendix, as indicated in the manuscript. Hence, we have attached it in this revision as S1 Appendix A. Also, we have included adequate information on the measure of workplace bullying, as suggested. Thank you for your input.

Note: 

Also, we have duly addressed the issue of the acronym WPB as it is used in the study i.e.

Workplace bullying (WPB)

---

## [Editor Report · Decision Letter 2]

16 Oct 2024

Investigating workplace bullying, intention to quit and depression among nurses in the Upper West Region of Ghana.

PONE-D-24-20237R2

Dear Dr. Dapilah,

We’re pleased to inform you that your manuscript has been judged scientifically suitable for publication and will be formally accepted for publication once it meets all outstanding technical requirements.

Kind regards,

Francesco Marcatto, Ph.D.

Academic Editor

PLOS ONE
---

## [Editor Report · Acceptance letter]

18 Oct 2024

PONE-D-24-20237R2 

PLOS ONE

Dear Dr. Dapilah, 

I'm pleased to inform you that your manuscript has been deemed suitable for publication in PLOS ONE. Congratulations! Your manuscript is now being handed over to our production team.

Kind regards, 

on behalf of

Dr. Francesco Marcatto 

Academic Editor

PLOS ONE